# A Single-Component Blue Light-Induced System Based on EL222 in *Yarrowia lipolytica*

**DOI:** 10.3390/ijms23116344

**Published:** 2022-06-06

**Authors:** Zhiqian Wang, Yunjun Yan, Houjin Zhang

**Affiliations:** Department of Biotechnology, College of Life Science and Technology, Huazhong University of Science and Technology, MOE Key Laboratory of Molecular Biophysics, Wuhan 430074, China; d201880529@hust.edu.com (Z.W.); yanyunjun@hust.edu.cn (Y.Y.)

**Keywords:** *Yarrowia lipolytica*, blue light induction, optogenetic, gene expression

## Abstract

Optogenetics has the advantages of a fast response time, reversibility, and high spatial and temporal resolution, which make it desirable in the metabolic engineering of chassis cells. In this study, a light-induced expression system of *Yarrowia lipolytica* was constructed, which successfully achieved the synthesis and functional verification of Bleomycin resistance protein (BleoR). The core of the blue light-induced system, the light-responsive element (TF), is constructed based on the blue photosensitive protein EL222 and the transcription activator VP16. The results show that the light-induced sensor based on TF, upstream activation sequence (C120)_5_, and minimal promoter CYC_102_ can respond to blue light and initiate the expression of GFPMut3 report gene. With four copies of the responsive promoter and reporter gene assembled, they can produce a 128.5-fold higher fluorescent signal than that under dark conditions after 8 h of induction. The effects of light dose and periodicity on this system were investigated, which proved that the system has good spatial and temporal controllability. On this basis, the light-controlled system was used for the synthesis of BleoR to realize the expression and verification of functional protein. These results demonstrated that this system has the potential for the transcriptional regulation of target genes, construction of large-scale synthetic networks, and overproduction of the desired product.

## 1. Introduction

Efficient control of cell production of target products is one of the main goals of developing regulatory elements [1,2,3]. Optogenetic tools have become important control elements for regulating various biological systems [4,5,6]. Recently, many optogenetic systems have been reported in the field of synthetic biology, especially several light-oxygen-voltage (LOV)-type photoreceptors [7,8] such as VVD [9], AsLOV2 [10], EL222 [11], etc. Their small size and endogenous cofactors, such as flavin adenine dinucleotide (FAD) and flavin mononucleotide (FMN) make LOV-type photoreceptors to be ideal candidates for developing novel optogenetic tools [12,13].

The light regulatory transgene system has the advantages of easy control, high induction efficiency, and low leakage [14], which is a potential tool for functional genome research, drug discovery, and gene therapy [15]. Chemically inducible gene expression systems have been widely used to efficiently control gene expression over the past few decades, and several systems have been commercialized. However, the toxicity [16], off-target effects [17], and non-specificity [18] of chemical inducers limit the widespread application of these systems. Compared with chemical substances, light has the advantages of low toxicity, easy availability [19], easy manipulation [20], and high spatial and temporal resolution [21], making it an ideal inducer that can achieve spatiotemporal control of transgene expression.

For example, the yLightOn system [22] developed by Yang. et al. based on VVD can be effectively applied to the blue light control cell growth and cell cycle of *Saccharomyces cerevisiae*; the OptoEXP system [23] developed by Avolos. et al. based on EL222 can be applied to the blue light-induced production of isobutanol in *S. cerevisiae*; The LOV-TAP system [24] developed by Sosnick. et al. based on *As*LOV2 and Trp suppressor can be applied to the blue light suppression regulation of *Escherichia coli*. Ye. et al. had developed a green light-inhibiting system for *Y. lipolytica* with the fusion protein CarH-VPRH as the core photosensitive element [25]. We previously reported a simple and robust light-switchable gene expression system for *Pichia pastoris* [26].

As an unconventional oleaginous yeast, *Y. lipolytica* has passed the ‘Generally Recognized as Safe’ (GRAS) certification of Food and Drug Administration (FDA) and can be used in the production of food and medicine [27,28]. *Y. lipolytica* has many natural lipid metabolism pathways and strong protein secretion ability. It is a kind of chassis cell that has been widely studied and used in recent years [29,30,31]. However, the results of whole-genome sequencing analysis of *Y. lipolytica* have a lower similarity of protein-coding genes compared with *S. cerevisiae*, so many difficulties have arisen in the development of genetic manipulation tools for *Y. lipolytica* [32,33,34]. At present, *Y. lipolytica* has successfully developed fatty acid and alkane inducible promoters [35], ethanol inducible promoters [36] and xylose inducible sensors [37] and other regulatory elements [38]. Among the reported systems, the erythritol inducible systems are one of the best currently constructed for *Y. lipolytica*, which can serve as versatile inducible promoters in *Y. lipolytica* [39,40,41]. However, these regulatory elements belong to traditional exogenous chemical inducers and soluble compounds, which still have problems such as transport delay, toxicity, and irreversibility. Optogenetic tools can compensate for these shortcomings to a certain extent.

EL222 from *Erythrobacter litoralis* is a LOV type photosensitive protein [42]. The blue light-induced rearrangement in the EL222 leads to dimerization by unmasking the C-terminal helix-turn-helix DNA binding effector from the surface of the photosensory LOV domain [42,43,44]. The light regulation system based on EL222 have been verified in *E. coli* [45], *S. cerevisiae* [46], *P. pastoris* [26] and zebrafish [47]. In the current research, we are committed to developing a compact and stable blue light induction system suitable for *Y. lipolytica* based on a similar design.

In this study, we developed a blue light induction gene expression system used in *Y. lipolytica* and named pYLBI (*Yarrowia lipolytica* Blue light Induction system). We show that the blue-light activated transcription factor VP16-EL222 [48] works in *Y. lipolytica* with the help of a customized promoter. The pYLBI system also shows 128.5-fold induction rate, reversibility, rapid response, and high temporal and spatial resolution. In addition, when used as the BleoR reporter gene, we can observe the acquisition of bleomycin resistance, thus broadening the toolbox for the development of light-inducible elements of *Y. lipolytica*.

## 2. Results

### 2.1. Design, Optimization and Verification of Blue Light Induction System Based on EL222

Based on the existing research work on *S. cerevisiae* and *P. pastoris*, we can learn from the eukaryotic design method of the EL222 system [26,49]. We constructed a single-component light-inducible protein composed of nuclear localization signal SV40 [50], transcription activation domain VP16 [51] and light-sensitive response protein EL222 named SVEL (SV40-VP16-EL222). In theory, SVEL can achieve light-induced homodimerization and bind to the upstream activation sequence C120 [43] to activate transcription (Figure 1a). To this end, we constructed a response fragment with GFPMut3 as the reporter gene, which is controlled by 5 copies of C120 ((C120)_5_) and a minimal eukaryotic promoter. An activation fragment composed of the promoter *hp4d* [52], its driven SVEL and *xpr2* terminator [53] was constructed upstream of the response fragment. In the final construction, these two fragments are located on one plasmid (Figure 1b).

To fine-tune the induction characteristics of the pYLBI system, we used a series of minimal promoters. Plasmids with different minimal promoters driving the GFPMut3 gene were introduced into Po1g cells (Appendix A). A variety of regulatory systems with different levels of leakage expression and maximum activation were observed after being induced with blue light for 8 h (Figure 1c). Among these minimal promoters, CYC_102_ [54], GAP_150_ [55] and TEF_136_ [56] achieved better light-inducibility and lower leakage expression, and CYC_102_ had the best effect, achieving 45-fold induction (Figure 1c).

Next, we changed the C120 repeat number from one to six copies. Our results showed that the expression level of GFP increased with the increase of C120 copy number, reaching the maximum induction rate at five copies (Figure 1d). These results indicate that the performance of the pYLBI system can be fine-tuned by changing the configuration of the promoter, which provides a flexible choice for specific experimental conditions.

### 2.2. Multi-Copy Construction and Verification of Response Fragments

By introducing *Xba*I and *Nhe*I restriction sites upstream and downstream of the response fragment, the isocaudamer method [57] can be used to construct multiple copies of the response fragment (Figure 2a). We chose the response fragment composed of (C120)_5_, CYC_102_, GFP and *xpr2* terminator for multi-copy construction and named the single response fragment 1 × CYC. The constructed 1 × CYC, 2 × CYC, 3 × CYC, and 4 × CYC were respectively integrated into Po1g and placed in a blue light illumination to induce 8h, and the induction intensity of transformed cells was quantified by measuring the fluorescence intensity of GFPMut3 (Figure 2b). Our results showed that the expression level of GFP increased with the increase of CYC copy number, and the induction rate reached 128.5-fold when the response fragment reached four copies. The results of flow cytometry also proved the above conclusions (Figure 2c).

### 2.3. Dose-Dependent Activation

In order to further verify whether the blue light irradiation time affects the expression level of the pYLBI system, we cultured Po1g cells carrying 1~4 × CYC plasmids in the dark to the early exponential phase and then induced them with full blue light for 10 h. Cultivation samples are taken every hour to detect the average fluorescence intensity. As the blue light irradiation time increases, the fluorescence intensity also changes. The results showed that all four groups reached the peak value after blue light induction for 8 h (Figure 3a, Appendix A).

A key advantage of a light-switchable promoter system lies in the ease of tuning gene expression levels by modifying illumination protocols. We designed a 10/100 s Blue, 30/100 s Blue, 50/100 s Blue, 70/100 s Blue, 90/100 s Blue with the same intensity of blue light emitted by a periodic light source based on the relevant situation reported by Avalos [23]. The results prove that there is a positive correlation between the induction intensity of pYLBI, the periodic supply of blue light and the copy number of the response fragment CYC (Figure 3b,c).

Using constitutive promoters (pGAP, pTEF) as the positive control group, we compared the highest fluorescence intensity obtained after 8 h of induction of the 4 × CYC circuit with the two positive control groups (Figure 3d). The highest expression level of 4 × CYC is 66% ± 3% of pGAP level and 45% ± 1% of pTEF level under the same all-blue culture conditions.

### 2.4. Assess the Performance at the Single-Cell Level

To further verify whether the blue light irradiation time affects the expression level of the pYLBI system, we use flow cytometry and fluorescent confocal technology to evaluate the performance of pYLBI at the single-cell level. Po1g cells carrying pYLBI series of plasmids were cultured in all Blue, 10/100 s Blue, 30/100 s Blue, 50/100 s Blue, 70/100 s Blue, 90/100 s Blue, or Dark. Although the detection limits of flow cytometry and microplate readers are different, our data shows that most cells respond as expected. Notably, blue light switches pYLBI in an all-or-none fashion with a small overlap in the extent of single-cell fluorescence in the dark and under all blue (Figure 4a), and the confocal fluorescence images (Figure 4b) show that the Po1g cells of five experimental groups had the high expression of GFP at the single-cell level.

### 2.5. Light Switchable Spatial Activation

Compared with chemical regulation, optogenetic regulation has the advantage of precise spatial control. Good spatial control enables the pYLBI system to induce cell expression at designated locations, while traditional chemical control can only achieve overall control. In order to control gene expression spatially, engineered cells transformed with pYLBI system plasmid with GFPMut3 as the reporter gene were grown on a solid medium and irradiated with blue light using a mask with a specific image. The GFPMut3 fluorescence image of the cell has the pattern of the original image used as a mask (Figure 5). After 10 h of induction, the expression of GFP was clearly visible in the illuminated area, and there was no fluorescence reaction in the shielded area. The GFP expression pattern on the plate exactly matches the shape of the photomask, indicating that the optogenetic system has achieved effective spatial control.

### 2.6. Temporal Behavior of pYLBI System

To study the temporal behavior of our system, we measured the GFP expression strength of Po1g carrying pYLBI systems in a 2-h “ON-OFF-ON-OFF-ON” cycle for 10 h. Every “ON” (100% lighting pulse) cycle has 2 h of all blue illumination. The culture was grown in the presence or absence of the blue light, thus exhibiting activation or inactivation. Our data shows that during the first 2-h “ON” period, the ΔGFP (ΔGFP = GFP_(x+1)h_ – GFP_xh_) in this system shows an upward trend, which is positively correlated with the induction time. During the subsequent 2-h “OFF” cycle, ΔGFP rapidly decreased and approached the inactivated state. The subsequent “ON-OFF-ON” cycle also showed this trend and the ΔGFP value in the induced state gradually increased (Figure 6a).

To test how fast the system could respond to blue illumination, we used qPCR to measure the mRNA level of GFP. After a 10-min cycle of “ON-OFF-ON-OFF-ON” treatment, we used the normalized cycle threshold (ΔC_T_) (ΔC_T_ = C_T_GFP-C_T_actin) to quantify the relative expression level in each sample (Figure 6b). As expected, the ΔC_T_ for blue-light treated 1~4 × CYC samples decreased with the increase of CYC copy number. Then we analyzed the increment of ΔC_T_ every 10 min, ΔΔC_T_ (ΔΔC_T_ = ΔC_T(x+10)min_ – ΔC_Txmin_). After 10 min of light, ΔΔC_T_ showed a significant decrease, and the degree of decrease was positively correlated with the number of CYC copies. After switching into the dark for 10 min, the value of ΔΔC_T_ tended to 0, indicating that the mRNA level of GFP did not increase significantly in the dark environment. Through the verification of qPCR, we have shortened the response time of the pYLBI system to the minute level. As a gene expression system with a high spatial and temporal resolution, pYLBI can be used to analyze the expression patterns of important genes at the single-cell and minute levels.

### 2.7. Blue Light-Induced Antibiotic Resistance

Due to its low background activity and high induction rate, pYLBI is suitable for strict control of the expression of target genes. *Y. lipolytica* is naturally tolerant to most commonly used antibiotics. Only a few antibiotics, such as bleomycin, hygromycin, and phleomycin, families, can be used as selection markers. We have demonstrated the antibiotic resistance marker Bleomycin resistance protein (BleoR) can be induced with blue light (Figure 7).

The GFPMut3 reporter gene in the 1 × CYC plasmid was replaced with the BleoR gene by the In-Fusion Cloning to obtain a single-copy blue light-induced BleoR expression plasmid. The subsequent construction of multi-copy plasmids was carried out by isocaudamer method. Under dark, non-inductive conditions, 50~250 μg/mL bleomycin can completely inhibit or delay the growth of Po1g cultures containing 1~4 × CYC. Incubation in constant blue light can induce BleoR expression and promote growth even in the presence of high concentrations of antibiotics. With the increase of the copy number of the blue light response fragment CYC, the concentration of bleomycin that Po1g cells can tolerate is higher.

## 3. Discussion

In recent years, there have been great developments in optogenetics regarding the tools and application fields. We have demonstrated the construction and characterization of pYLBI, a single-component blue light-induced system based on EL222 in *Y. lipolytica*. Our research expands the optogenetics in *Y. lipolytica*, which has the potential to add a new control layer to *Y. lipolytica* fermentations. We anticipate that these designs and methods will facilitate the development of new tools for *Y. lipolytica*.

The model organisms such as *S. cerevisiae* and *E. coli* have a well-established synthetic biology toolbox due to their well-characterized, rapid growth and ease of stable transformation [58,59,60]. Other unconventional yeasts have many unique metabolic characteristics, such as the utilization of special carbon sources, metabolism of specific substances, high-density culture, and special secretion mechanisms, which make them more competitive in industrial applications [61,62]. With the development of new synthetic biology tools, some previously unconventional yeasts, such as *Y. lipolytica* and *P. pastoris*, are becoming model organisms [63,64].

Here, we demonstrate the development and engineering of a synthetic blue-light-inducible regulatory system, pYLBI, suitable for *Y. lipolytica*. We have proved that pYLBI can serve as an excellent scaffold for the development of novel inducible metabolic pathways that can control the on and off of target protein synthesis temporally and spatially. Then we studied the effects of a series of parameters such as the number of upstream activating sequence C120, the type of core promoters and the copy number of the response fragment on the regulation effect of blue light, and achieved an a higher on/off ratio induction ratio (>128.5-fold) in *Y. lipolytica* than in *P. pastoris* (79.7-fold) for only 8-h induction, with extremely low leakage and high activation efficiency (66% ± 3% of the pGAP level and 45% ± 1% of the pTEF level). The previously reported green light regulation system based on CarH-VPRH for *Y. lipolytica* could only achieve 43-fold induction after continuous induction for 72 h [25].

Modularity is a concept widely used in biological sciences, especially in synthetic biology research. Compared with the CarH-VPRH optogenetic system, which requires fusion expression of CarH, VP64, p65, Rta and HSF1 to function [37], the components of the pYLBI system are more streamlined, which can effectively reduce the exogenous gene pressure of cells and improve its applicability in the construction of large-scale metabolic pathways. Unlike the CarH-VPRH system, which requires additional exogenous cofactor AdoB12 [25], pYLBI system does not need to introduce any exogenous chromophores and only relies on the abundant FMN in the medium as a cofactor to achieve its function. Moreover, since the core photosensitive protein SVEL of the pYLBI system has a smaller molecular weight (32.3 kDa) than CarH-VPRH (100.4 kDa) [37,65], SVEL can achieve better modularity and rapid synthesis and response. Smaller molecular weights also avoid steric hindrance and greatly facilitate the precise molecular design of optogenetic tools.

Rapid response is one of the characteristics of an ideal gene-inducible expression system. On the protein reporter expression level, the t_1/2_ (the time to reach half of the max expression) of the pYLBI system was ~1.1 h in all of the 1~4 copy systems (Figure 6a), while the t_1/2_ of the LVAD (Y50W/M135I), CRY2/CIB1, yLightOn are ~2.6 h, ~4.7 h, and ~1.8 h, respectively [22]. Concerning the mRNA-level response, pYLBI could achieve response and reversal under short-term light (10 min) and dark cycles, which demonstrates that our pYLBI system performs well in dose-response behavior and deactivation kinetics.

## 4. Materials and Methods

### 4.1. Assembly of DNA Bioblocks

For the construction of all plasmids, in-fusion cloning was performed according to standard molecular biology techniques. The principle is to intercept the main part of the known eukaryotic constitutive promoter and add the DNA binding site of EL222 upstream to achieve the purpose of blue light-induced expression.

According to the above principle, a simple, compact and blue-light induced plasmid can be obtained. The backbone of the pYLBI plasmids is derived from the pINA1296 plasmid (*hp4d*/*XPR2* pre/-/*XPR2t*, expression cassette; *LEU2*, selection marker; pBR322, integration site.) [66] to ensure compatibility with existing protocols. All plasmids were designed using Snapgene (GSL Biotech LLC., San Diego, CA, USA) and constructed by GenScript Biotech (Nanjing, China) with In-Fusion Cloning. The *hp4d* promoter, minP_64_, TRP_148_, HIS3_188_, CYC_102_, GAP_150_, TEF_136_, *xpr2* terminator, SVEL, GFPMut3 and BleoR genes were synthesized as bioblocks.

Firstly, the SVEL gene was integrated downstream of the *hp4d* promoter with In-Fusion Cloning, thus achieving the activation fragment composed of *hp4d*, SVEL, and *xpr2* terminator. Then, the construction of pYLBI plasmid can be completed by integrating the response fragment composed of (C120)_5_, core promoter (minP_64_, TRP_148_, HIS3_188_, CYC_102_, GAP_150_, TEF_136_) and *xpr2* terminator into the downstream of the activation fragment. Subsequently, the reporter gene GFPMut3 can be replaced by BleoR to realize the change of gene of interest. Finally, a plasmid with multiple copies of the response fragment can be constructed using the isocaudamer method.

### 4.2. Strains and Growth Conditions

The *Y. lipolytica* Po1g strain (*MATa*, *leu2–270, ura3–302::URA3, xpr2*–322, axp1–2, pXPR2-SUC2 derived from Po1f, both extracellular proteases deleted, grown on sucrose, pBR322 docking platform, Leu^−^) [52] was used as the host for the functional verification of the pYLBI system. All characterization experiments were performed using MD medium (1 L), including 13.4 g yeast nitrogen base as nitrogen source, 20 g glucose as carbon source and 0.4 mg biotin. Unless otherwise specified, liquid MD medium was cultured in a 250 mL Erlenmeyer flask at 28 °C and 200 rpm, and solid yeast was cultured at 28 °C. Seed cultures from freshly transformed plates were grown overnight in 5 mL MD medium.

### 4.3. Microplate Reader Measurements

Use FlexStation 3 (Molecular Devices, LLC., San Jose, CA, USA) to measure GFP fluorescence (excitation wavelength 488 nm; emission wavelength 525 nm) and optical density 600 nm (OD_600_). The optimal gain was used and kept through all measurements. Then, the GFP/OD_600_ value of cells lacking the GFP structure was subtracted from the fluorescence value (GFP/OD_600_) of each sample for correction. Therefore, the reported value is calculated according to the following formula, where GFP corresponds to fluorescence measurement and OD corresponds to OD_600_.
GFP/OD_Strain,Conditon_ = (GFP_Strain,Conditon_ – GFP_Media,Conditon_)/(OD_Strain,Conditon_ – OD_Media,Conditon_) – (GFP_No GFP Control Strain,Conditon_ – GFP_Media,Conditon_)/(OD_No GFP Control Strain,Conditon_ – OD_Media,Conditon_)

Unless otherwise described, all fluorescence measurements were carried out at the end of the experiment or on cultured samples so that the potential activation of VP16-EL222 by the light used to stimulate GFP did not affect our experiments or results.

### 4.4. Flow Cytometry

The Po1g cells carrying the pYLBI system grown under non-induced conditions to the early exponential phase were cultured for 8 h under a series of blue illumination conditions. 5 × 10^4^ cells were analyzed on CytoFLEX (Beckman Coulter, Inc., Indianapolis, IN, USA). All parameters remain unchanged to allow direct comparison of structures and conditions. Signals arising from viable cells were sorted out from those originating from aggregates and debris based on forward and sideward scattering signals using FlowJo (Tree Star Inc., Ashland, OR, USA). Use Origin 2019b (OriginLab Co. Ltd., Northampton, MA, USA) to analyze the data.

### 4.5. Confocal Fluorescence Imaging

The Po1g cells carrying pYLBI series plasmids have been induced to the maximum fluorescence intensity under all blue. The cells were then collected by centrifugation and resuspended in PBS. Use FV1000 laser scanning confocal microscope (Olympus, Tokyo, Japan) and a 60× objective lens to capture fluorescence images.

### 4.6. Imaging of Agar Plates

The Po1g culture, which transformed CYC_102_ plasmid and the soft agar MD solution (7 g/L agar) are evenly mixed and poured into a 90 mm petri dish. The inverted plate is covered with a designed photomask, and then placed under parallel blue light irradiation for 10 h. The photos were taken through the orange transparent filter. The photomask was designed with Autocad 2019 (Autodesk, Inc., SanRafael, CA, USA) and made with film by Kaisheng Electronic Technology Co., Ltd. (Suzhou, China).

### 4.7. Quantitative PCR (qPCR) Analysis

Total RNA was isolated from target Polg cells using an E.Z.N.A Yeast RNA Kit (Omega Bio-tek, Inc., Norcross, GA, USA) following the manufacturer’s instructions. qPCR was performed on Applied Biosystems™ QuantStudio™ 3 (Thermo Fisher Scientific Inc., Waltham, MA, USA). Samples were run in triplicate and the average cycle threshold (C_T_) was calculated.

## 5. Conclusions

Overall, we developed a blue light-activated gene expression system for *Y. lipolytica*. As a new member of the synthetic biology toolbox, pYLBI possesses many favorable inducible properties, such as one-component construction, direct activation, highly tunable inducible properties, precise spatiotemporal resolution and good adaptability, which can achieve rapid, reversible, quantitative, and spatiotemporal control of gene expression in *Y. lipolytica*. This system is expected to act as a convenient tool for studying gene regulatory networks in unconventional yeast and for the large-scale production of recombinant proteins.

## Figures and Tables

**Figure 1 ijms-23-06344-f001:**
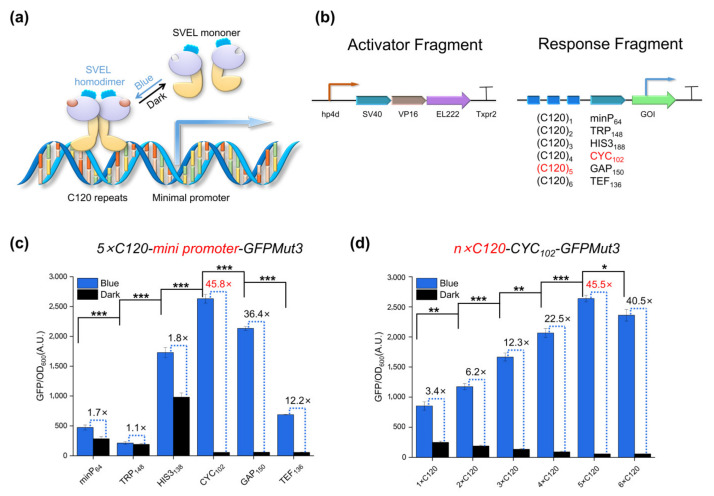
Design, optimization and validation of the single-component optogenetic system. (**a**) Blue light induction mechanism of EL222. (**b**) Configuration of the single-component light-switchable gene expression system. (**c**,**d**) Fine tuning pYLBI system by altering the promoter configurations, including different minimal promoters (**c**) and the C120 copy number from 1 to 6 (**d**). Unless otherwise stated, error bars indicate Means ± S.D. (n = 3); Significance of differences between all blue induced states: * *p* < 0.05, ** *p* < 0.01, *** *p* < 0.001.

**Figure 2 ijms-23-06344-f002:**
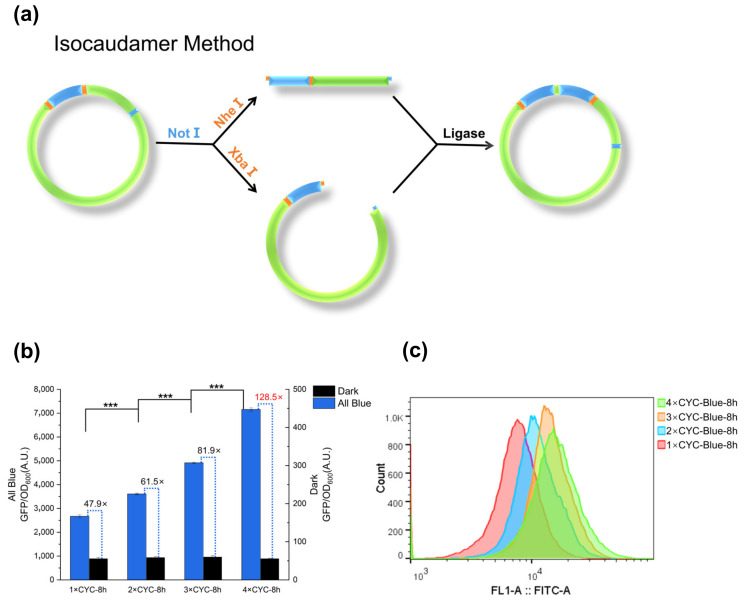
Design, optimization and validation of the single-component optogenetic system. (**a**) Construction of multiple copies of response fragments by the isocaudamer method. The cyan part represents the response fragment. (**b**) Strains harboring response fragments with different copies from 1 to 4 (1~4 × CYC) were cultured for 8 h with or without blue light. (**c**) Flow cytometry detection of strains carrying different plasmids after induced to the highest fluorescence intensity under blue light. Error bars indicate Means ± S.D. (n = 3); Significance of differences between all blue induced states: *** *p* < 0.001.

**Figure 3 ijms-23-06344-f003:**
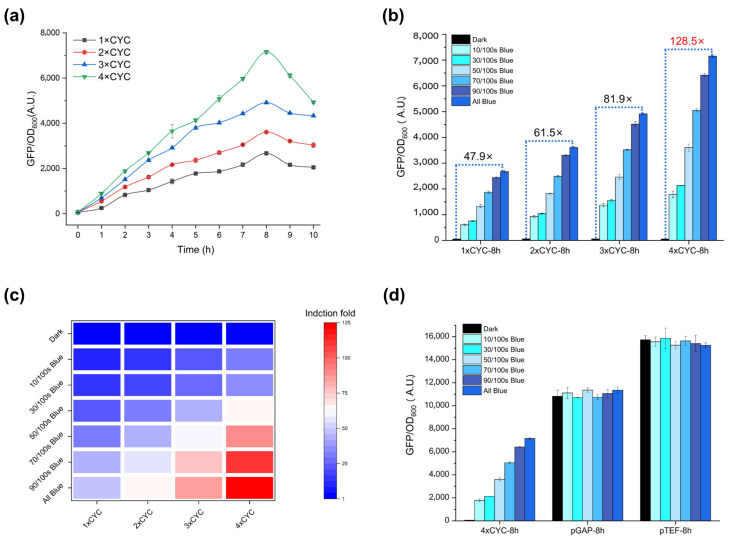
Dose-dependent activation. (**a**) 1~4 × CYC pYLBI circuits induced under constant blue light for 10 h. All groups reached the maximum fluorescence intensity after 8-h induction. (**b**) The Po1g harboring 1~4 × CYC pYLBI plasmids were induced for 8 h under a series of blue illumination conditions. (**c**) Dose-dependent activation with different illumination pulse and CYC copies after 8-h induction. (**d**) FI of 4 × CYC, pGAP and pTEF incubated under different periodic blue light conditions for 8 h. Under all-blue culture conditions, 4 × CYC can reach 66% ± 3% of pGAP level and 45% ± 1% of pTEF level. Error bars indicate Means ± S.D. (n = 3).

**Figure 4 ijms-23-06344-f004:**
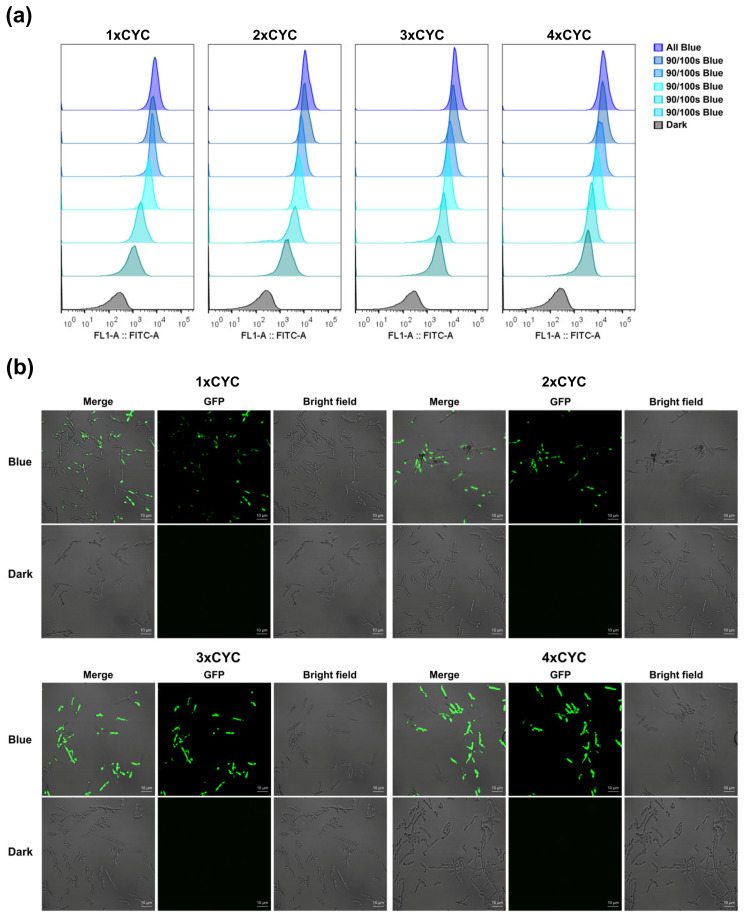
Assessment of the performance at the single-cell level. (**a**) Flow cytometry of Po1g cells harboring 1~4 × CYC pYLBI circuits. All of the cells were induced to the maximum. Flow cytometry images of every group are sorted as All Blue, 90/100 s Blue, 70/100 s Blue, 50/100 s Blue, 30/100 s Blue, 10/100 s Blue, and Dark from top to bottom. A total of 5 × 10^4^ cells were analyzed for each population. The results acquired by analyzing the data from flow cytometry are consistent with the data obtained by the microplate reader. (**b**) The confocal images of All blue and Dark groups. From left to right are the Merge group, GFP group, and Bright field group, respectively. The gain of every image remains consistent. The single cells of all blue groups showed strong GFP signals. There was no GFP signal in the dark groups. Scale bars represent 10 µm. All above-mentioned experiments verify thepYLBI systems have a strong response to blue light.

**Figure 5 ijms-23-06344-f005:**
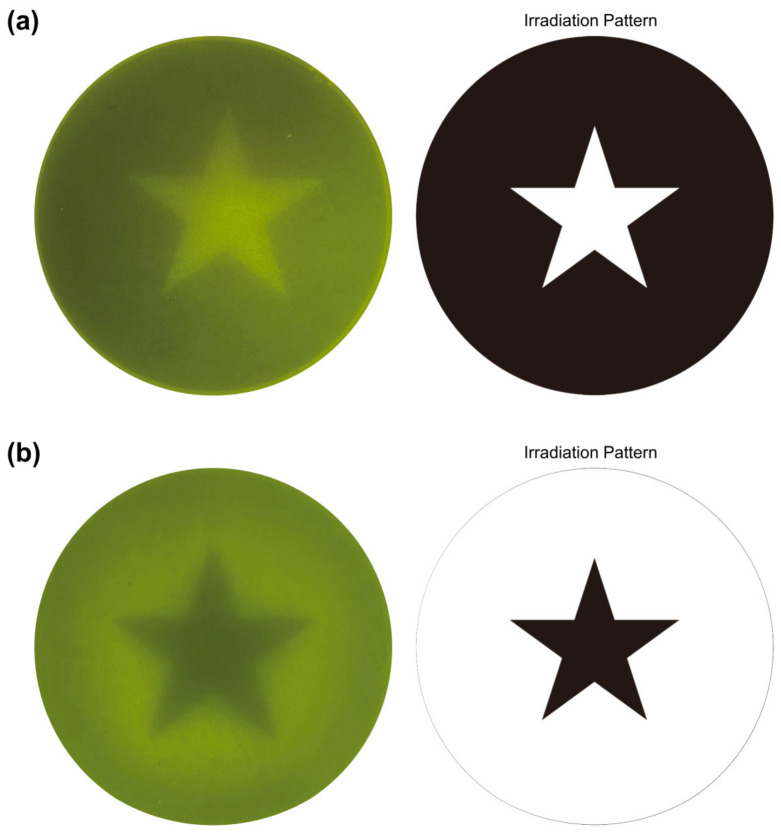
Light switchable spatial activation. Cells transformed with the 1 × CYC plasmid were plated on MD agar plates and illuminated with blue light for 10 h through a photomask with a star-shaped shield. The fluorescent pattern on the agar plate could precisely reproduce the pattern on the photomask, showing that the optogenetic system has a good degree of spatial control. The masks used for (**a**,**b**) are completely complementary.

**Figure 6 ijms-23-06344-f006:**
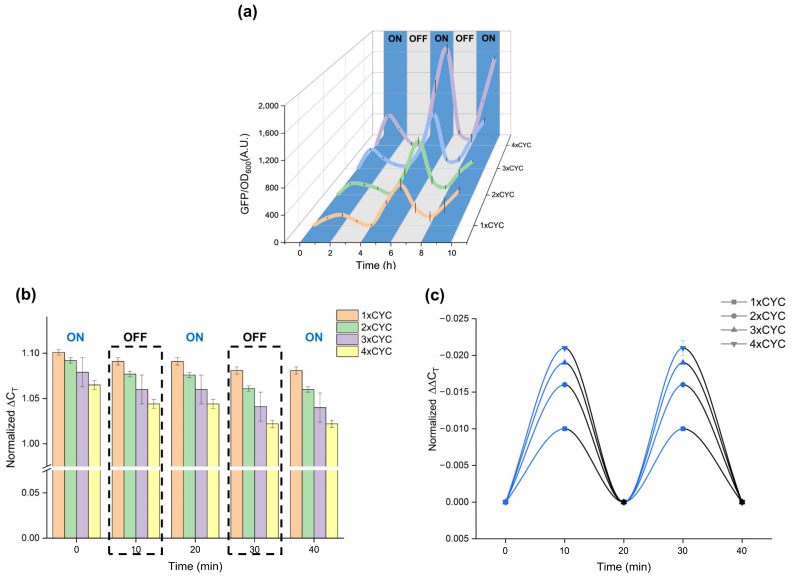
Temporal behavior of pYLBI system. (**a**) GFPMut3 activity levels in 1~4×CYC samples were illuminated or kept in the dark for the indicated times (2 h). Grey areas represent the dark state (‘OFF’) while the blue areas (illumination ‘ON’, all blue). (**b**) GFPMut3 mRNA levels were quantified by qPCR from 1~4 × CYC samples treated with blue light or kept in the dark with an on-off cycle of 10 min intervals. Normalized ΔC_T_ values are shown. (**c**) Normalized ΔΔC_T_ value processed for 10 min in the activated (Blue line) or deactivated (Dark line) state. Error bars indicate Means ± S.D. (n = 3).

**Figure 7 ijms-23-06344-f007:**
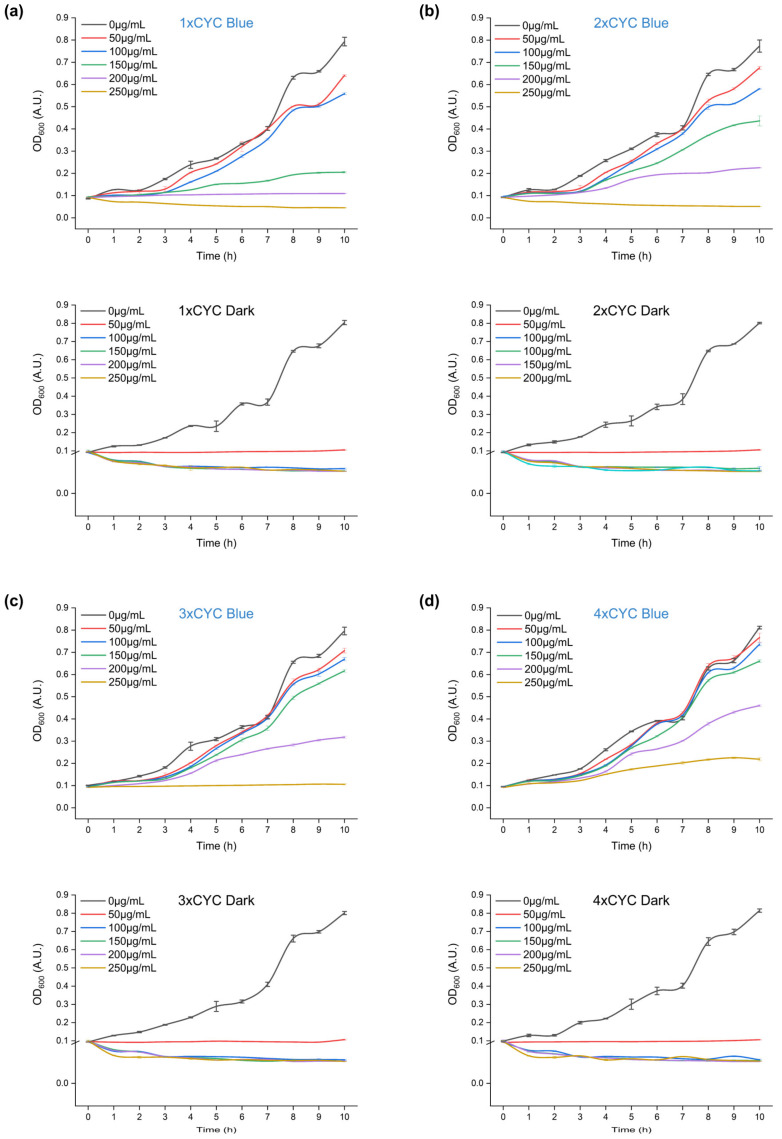
Blue light-induced antibiotic resistance. (**a**–**d**) show the growth profile of Po1g cells harboring 1~4 × CYC plasmids under bleomycin concentration gradient, respectively. In sub-panels, the blue title represents the blue light-induced experimental group, and the black title represents the dark-cultured control group. Hence, incubation in constant blue light induces BleoR expression and facilitates growth even in the presence of high concentrations of the antibiotic. Error bars indicate means ± S.D. (n = 3).

## Data Availability

All are presented in the article or available in raw form from the corresponding author upon reasonable request.

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
