# Peer review of "A Single-Component Blue Light-Induced System Based on EL222 in Yarrowia lipolytica"

_ijms, 2022, doi:10.3390/ijms23116344_

Round 1
Reviewer 1 Report
The Authors developed a light-induced expression system operating in Y. lipolytica. While the subject, approach and scope of experimentation are interesting and definitely wort publishing, the quality of article is very poor. I’d like to ask the Authors to correct the manuscript, so that it could be actually published, as the present form of text formatting and data presentation it is unacceptable.
In my opinion, the Article is very chaotic. For example, considering only the Introduction:
1) the same threats are repeated several times (L 28 -30 and again L 42 – 44; or L 30 – 33 and again L 44 – 45 – these are only examples –such repetitions appear in many spots lines
2) some information is misleading, which stems more from awkward style, like
2.a. “ The light regulatory transgene system …… which is an indispensable tool for…” – I cannot agree with this, as gene expression can be induced by the other methods as well, so the system Is not “indispensable”
2.b “Y. lipolytica has a low level of post-translational glycosylation “ – this is not true
2.c “Broadened the toolbox for the development of light-inducible elements of Y. lipolytica” – this is not a grammatically correct sentence
the same in L 162 :”To further verify whether the blue light irradiation time affects the expression level of the pYLBI system”
the same in : 234 “We have demonstrated the antibiotic resistance marker Bleomycin resistance protein (BleoR) (Figure 7).” – this is not a complete sentence.
2.d the text is sometimes written in past and sometimes in the present tense, which is difficult to read
L 65 – please mention erythritol inducible systems, which are one of the best from currently constructed for Y. lipolytica (Trassaert et al. 2017; Park et al. 2019b; Park et al. 2019a).
Park Y-K, Korpys P, Kubiak M, Celińska E, Soudier P, Trébulle P, Larroude M, Rossignol T, Nicaud J (2019a) Engineering the architecture of erythritol-inducible promoters for regulated and enhanced gene expression in Yarrowia lipolytica. FEMS Yeast Res 19:1. doi: 10.1093/femsyr/foy105
Park YK, Vandermies M, Soudier P, Telek S, Thomas S, Nicaud JM, Fickers P (2019b) Efficient expression vectors and host strain for the production of recombinant proteins by Yarrowia lipolytica in process conditions. Microb Cell Fact 18:167. doi: 10.1186/s12934-019-1218-6
Trassaert M, Vandermies M, Carly F, Denies O, Thomas S, Fickers P, Nicaud JM (2017) New inducible promoter for gene expression and synthetic biology in Yarrowia lipolytica. Microb Cell Fact. doi: 10.1186/s12934-017-0755-0
L 93 – 96 – the style is illegible – please correct
I’m also not sure about correctness of referencing, for example – the reference for isocaudomers usage is supported by citation of : “He, W.; Mu, W.; Jiang, B.; Yan, X.; Zhang, T. Food-Grade Expression of d-Psicose 3-Epimerase with Tandem Repeat Genes in Bacillus subtilis. J Agric Food Chem 2016, 64, 5701-5707, doi:10.1021/acs.jafc.6b02209.”, which I find very weird.
L 121 – please define, what exactly 1xcyc, 2xcyc etc. refer to – that this is a complete expression cassette with a minimal cyc promoter
L 146 no reference to Avalos
L 159 – the caption states incorrect: The comparison of highest FI of 4×CYC circuits with 2 common constitutive promoters, pTEF (45%±1%) and pGAP (66%±3%) after 8-h induction.”. In fact the figure shows, that pTEF is non titratable, but the strongest promoter, and the numbers “(45%±1%) and (66%±3%) “ refer to the level of expression of 4xCYC and pGAP when compared to pTEF-driven expression level
L 161 – assess à assessment
Figure 4 caption does not state what are the three images from each treatment group (blue or dark)
Notation system must be improved and more precise – sometimes the Authros describe a plasmid as 1xCYC, the other time (L 195) as CYC102 plasmid. This must be consistent
L 201: “fluorescent expression of Po1g”? very imprecise…please correct
Fig.6. a. could be improved in quality – it is very difficult to see the trend lines.
The part on BleoR strain construction is completely unknown
Caption to Fig.7 –“growth situation “ – this must be changed; this is not a scientific term. Also, the caption does not explain what are the smaller graphics within the sub-panels a to d. Axes are not defined
L 247 – 248 – this is not true: “ However, other yeasts are generally better suited as biological hosts because their natural metabolism makes them more suitable for production targets of interest”, as it depends on the targeted molecule. Maybe state “are competitive”, but one cannot deifinitely state that S.c erevisiae and E coli are worse. This is not true
Discussion is chaotic – the introduction to discussion is in the last paragraph. Also, the current Discussion does not actually discuss nor interprets the obtained data. The results should be compared with the Authros system for P. pastoris and the Green-light-based system for Y. lipolytica.
Materials and Methods are insufficient to repeat the experiments. Primarily, a list of strains with, their phenotypes and genotypes is missing. It is not stated what are the properties of pINA1296 vector and what insertion site was used for Y. lipolytica Po1g. No details on construction assembly, sub-cloning, cloning,
Reviewer 2 Report
In the current manuscript, Wang et al. developed and applied a photoinducible system based on the photosensitive protein EL222 in Yarrowia lipolytica. The light-inducible transcriptional activating protein is a fusion of SV40 nuclear localization-VP16 activation domain and EL222. Upon photoinduction, the fusion protein dimerizes and binds to the Upstream Activation Sequence C120. The reporter gene GFPmut3 was expressed under the control of various minimal promoter elements as well as increasing the number of C120 UAS. Their setup was subsequently tested under different blue light regiments, and spatial and temporal activation. For functionality, induction of the bleomycin resistance gene was used.
Overall, the system was developed and applied well in Yarrowia, although in the temporal induction experiments the use of a destabized form of GFPmut3 would have been preferable, as the half-life of the protein is still long to see immediate induction events.
Issues to be addressed:
Some of the figures need to be reworked. Fig. 6a the blue color is too intense and the lines of the graphs too thin, and cannot be visualized well.
Fig 7. The graphs within graphs are too small and unintelligible. The a-d legends are not marked in the figure.
Changes in English usage: L155 constant blue light is more appropriate than “all blue”
L173 Change first sentence to Assessment of
L181 insert space
L252 correct error
